# An Upgrade on the Surveillance System of SARS-CoV-2: Deployment of New Methods for Genetic Inspection

**DOI:** 10.3390/ijms23063143

**Published:** 2022-03-15

**Authors:** José Francisco Muñoz-Valle, Alberto Antony Venancio-Landeros, Rocío Sánchez-Sánchez, Karen Reyes-Díaz, Byron Galindo-Ornelas, Wendy Susana Hérnandez-Monjaraz, Alejandra García-Ríos, Luis Fernando García-Ortega, Jorge Hernández-Bello, Marcela Peña-Rodríguez, Natali Vega-Magaña, Luis Delaye, Mauricio Díaz-Sánchez, Octavio Patricio García-González

**Affiliations:** 1Institute for Research in Biomedical Sciences (IICB), University Center for Health Sciences, University of Guadalajara, Guadalajara 44340, Mexico; drjosefranciscomv@cucs.udg.mx (J.F.M.-V.); jorge.hernandezbello@cucs.udg.mx (J.H.-B.); alejandra.vega@academicos.udg.mx (N.V.-M.); 2Translational Institute of Genomic Singularity (ITRASIG), Irapuato 36615, Mexico; antony.landeros@gmail.com; 3Molecular Design Department, Genes2Life (Grupo T), Irapuato 36615, Mexico; timina@grupot.xyz; 4Research and Development Department, Genes2Life (Grupo T), Irapuato 36615, Mexico; karenrdd03@gmail.com (K.R.-D.); tiempo_real@grupot.xyz (B.G.-O.); diagnostico@grupot.xyz (W.S.H.-M.); polimerasa@t4oligo.com (A.G.-R.); secuenciacion@t4oligo.com (M.D.-S.); 5Department of Genetic Engineering, Center for Research and Advanced Studies of the National Polytechnic Institute (CINVESTAV), Irapuato 36824, Mexico; luis.garcia@cinvestav.mx (L.F.G.-O.); luis.delaye@cinvestav.mx (L.D.); 6Laboratory for the Diagnosis of Emerging and Reemerging Diseases (LaDEER), University Center for Health Sciences, University of Guadalajara, Guadalajara 44340, Mexico; marcee24.p.r@gmail.com

**Keywords:** qPCR variant screening, SARS-CoV-2 variant identification, SARS-CoV-2 epidemiology, genetic surveillance

## Abstract

SARS-CoV-2 variants surveillance is a worldwide task that has been approached with techniques such as Next Generation Sequencing (NGS); however, this technology is not widely available in developing countries because of the lack of equipment and limited funding in science. An option is to deploy a RT-qPCR screening test which aids in the analysis of a higher number of samples, in a shorter time and at a lower cost. In this study, variants present in samples positive for SARS-CoV-2 were identified with a RT-qPCR mutation screening kit and were later confirmed by NGS. A sample with an abnormal result was found with the screening test, suggesting the simultaneous presence of two viral populations with different mutations. The DRAGEN Lineage analysis identified the Delta variant, but there was no information about the other three mutations previously detected. When the sequenced data was deeply analyzed, there were reads with differential mutation patterns, that could be identified and classified in terms of relative abundance, whereas only the dominant population was reported by DRAGEN software. Since most of the software developed to analyze SARS-CoV-2 sequences was aimed at obtaining the consensus sequence quickly, the information about viral populations within a sample is scarce. Here, we present a faster and deeper SARS-CoV-2 surveillance method, from RT-qPCR screening to NGS analysis.

## 1. Introduction

Since late 2019, coronavirus disease (COVID-19), an illness caused by a novel coronavirus called severe acute respiratory syndrome coronavirus 2 (SARS-CoV-2) has represented one of the main challenges of public health across the world. Along with the SARS-CoV-2 dissemination over new territories, new mutations such as the spike (S) protein mutation D614G (A23403G) emerged and became dominant over time [1,2,3]. After this evolutionary event, the population of non-D614G-mutants is virtually nonexistent, and it appears to be a consequence of the adaptation of the virus [4], but even after many studies, the reasons for this change in prevalent strains are not totally clear.

The S protein is characteristic of the coronavirus surface, and it is involved in the viral adsorption over the host cell surface because this protein interacts with the cellular receptors such as ACE2 (Angiotensin converting enzyme). Because of this, the S protein is one of the key molecules used as targets in COVID-19 vaccines [5,6]. Along with the replication and dissemination of the virus, several mutations arose and became fixated in the genome of SARS-CoV-2, originating variants of the virus. As variants diverge and accumulate mutations, it is expected that they have a heterogeneous epidemiological behavior, and in some cases even a differential clinical progression, but there is not enough data available to predict the result of mutations combined within a single viral particle [7].

Sampling, SARS-CoV-2 detection, and genetic analysis to identify genomic characteristics of infecting viruses are the major steps for epidemiological surveillance worldwide. However, there are important differences regarding these approaches: (i) the number of samples taken and assayed for the presence of SARS-CoV-2, (ii) data reported to corresponding Health Departments, (iii) criteria for sample selection as sequencing candidate, to list a few. Each government handles the situation as it appears to be the best option for their specific situation, but an essential aspect of this epidemiological approach is the economic situation. The price for virus detection by RT-qPCR has been reduced and become widely available, in contrast to sequencing technology. Moreover, it is important to note that NGS (Next-Generation Sequencing) requires different laboratory equipment, specially trained scientists, in addition to sequencing reagents, which makes the intensive use of NGS technology difficult in several countries. On the other hand, RT-qPCR technology is a readily available technology, and if it is correctly designed, it can help in the screening of samples for mutations. An affordable option of RT-qPCR technology for SARS-CoV-2 variant screening is Master Mut Kit (Genes2Life, Mexico), which can detect mutations present within the spike gene, and therefore, identify if the genetic material belongs to a VOI (Variant of Interest) or VOC (Variant of Concern) virus. As epidemiological surveillance becomes more scrupulous, data about the mutations and their real distribution will be available for most countries, and ultimately, it will have a higher certainty of epidemiological data accuracy. Additionally, as more tests are performed, now rare events, such as simultaneous infection with two or more strains of SARS-CoV-2 will become more frequently detected and relate to their actual occurrence.

NGS data analyses are commonly processed with public-available bioinformatics tools. As main programs and algorithms became widely used by researchers worldwide, the amount of genomic data generated each day increases substantially, representing a potential challenge because the processing power needed to supply the demand increases every day. Additionally, as the speed of sample analysis increases, the depth of analysis is reduced, therefore, losing important data, such as genetic populations. Some of the leading platforms for sequencing, such as ARTIC, obtain information of variants while processing the data, but this is performed at the last stage when a consensus sequence is obtained; all mutations below the threshold level for identification for the variant call are lost.

The threshold level of the Illumina DRAGEN COVID Pipeline is 0.5 (Illumina DRAGEN COVID Pipeline Software Guide, Document # 1000000158680 v01). This study aims to propose two methods for analyzing SARS-CoV-2, a RT-qPCR method that can accurately identify VOI and VOC at a lower cost and shorter time than NGS, and a bioinformatics data processing pipeline to obtain information from NGS reads which is currently lost in the regular analysis. Both objectives in order to demonstrate that the integration of both methodologies would make the current and future epidemiological surveillance programs and research protocols more efficient.

## 2. Results

### 2.1. Master Mut Analysis

Samples collected between March and October 2021 were analyzed with Master Mut Kit. Table 1 shows the summary of the variants found in the 87 samples that were analyzed.

Examples of RT-qPCR curves and the interpretation table from Master Mut Kit are available in Appendix A.

Undetermined samples are not VOI nor VOC, but this kit cannot determine their exact classification. The mutations present in them were: two samples with an absence of all mutations, one with just 69–70 deletion detected and the last one with R346K, L452R/Q, T478K, E484K and N501Y mutations. The sample code for this last one is M84. The Cq of each mutation was as follows: L452R/Q (Cq = 16.54), T478K (Cq = 16.57), E484K (Cq = 18.75), N501Y (Cq = 19.1), and R346K (Cq = 18.71), which suggest the mutations are not present in equal quantities, being L452R/Q and T478K more abundant than the other three. This result indicates the presence of the Delta variant as dominant, with the Mu variant as second. Amplification curves from this sample are available in Appendix A.

### 2.2. Concordance of SARS-CoV-2 Variant Identification by Master Mut and by Sequencing

All samples analyzed by Master Mut kit were further analyzed by sequencing with Illumina^®^ COVIDSeq™ Kit in an iSeq platform, and genome sequences were obtained with the Illumina DRAGEN COVID Lineage v3.5.3 app. Samples were prepared following manufacturer instructions. Fasta files were downloaded from the BaseSpace platform for further analysis.

The resulting SARS-CoV-2 genomes were identified using the Pangolin COVID-19 Lineage Assigner web application (Available at pangolin.cog-uk.io, last accession 14 December 2021). The resulting identifications were compared to the mutations and variants previously identified by the Master Mut kit.

For the four undetermined samples, which could not be identified with Master Mut, the identification was: Sample with 69–70 deletion (M34) belongs to B.1.1.222; samples without mutations belonged to B.1 (M40) and B.1.1 (M35). Sample M84 was identified as Delta.

Master Mut is capable of identifying VOI and VOC and distinguishing samples that did not belong to any of them. For 86 of 87 samples (98.5%), there was concordance between the Master Mut Kit analysis results and the data obtained from NGS sequencing with COVIDSeq Test. The only sample which did not have matching results between sequencing and Master Mut kit was M84 since the consensus sequence did not match all mutations previously described.

M84 sample presented five mutations in Master Mut Kit, but in the consensus sequence obtained from Illumina DRAGEN COVID Lineage v3.5.3 app, there were only two mutations, L452R and T478K, while R346K, E484K and N501Y were not present.

The fastq files of this and the other three samples were downloaded and analyzed locally.

### 2.3. Results from Local Data Analysis

Since a result from Illumina DRAGEN COVID Lineage v3.5.3 app was not fully concordant with the results from Master Mut (Sample M84), the sequencing reads from four samples (M81, M83, M84 and M86) were manually reviewed, mapped and assembled, in order to analyze and compare the information generated by NGS data processing tools, especially looking for data lost in simplification and automatic consensus generation. It was decided to analyze more samples than just M84 to test the procedure with samples apparently homogeneous, to decrease the possibility of misinterpretation of FreeBayes results. FreeBayes will analyze the mapped reads and calculate the relative abundance of mutations present, given a reference genome. With the parameters used in this paper, the groups will be 3; AC = 3 means the mutation is present in virtually all the reads, AC = 2 indicates the mutation is present in most reads, and AC = 1 indicates the mutation is present in few reads, but at least 15% of them.

#### 2.3.1. Sample M81

This sample was identified as a Delta variant. The mutations detected by DRAGEN are the same as detected by FreeBayes, except for GCT28086ACA, but it is important to notice that this mutation is grouped in AC = 1, which means its abundance is lower than 50%, to be exact, 150 reads have this mutation, while 226 have the wildtype allele; hence just a 40% of the reads present the mutation (Table 2). Since only 40% of the reads are mutated; therefore, the automatic analysis performed by the DRAGEN COVID Lineage app discards them.

#### 2.3.2. Sample M83

This sample was identified as a Lambda variant. FreeBayes (Table 3) detects three mutations not detected by DRAGEN, all of them are classified as AC = 1, of which 2 are near to the 3′ end of the viral genome. The mutation detected by FreeBayes at base 26,894 is to be noted, since 10,674 reads have it, while 15,404 reads had the native base, and the total depth at this position is 26,097, this means that although the mutation was detected in 40% of the reads, it was not represented in the result provided by DRAGEN. This synonymous mutation is located within the *M* gene of SARS-CoV-2.

Regarding the mutations near the 3′ end of the genome (C29370T and C29870A), the number of reads is very low compared with the rest of the genome. The reads at each position are 1427 and 54, respectively, and the abundance was below 25% of those reads. In contrast, the median depth was 4638. This mutation has been reported in other Lambda samples, but the low number of reads and their respective abundance, especially in the case of C29870A, difficulties the determination of mutation presence.

#### 2.3.3. Sample M86

This sample is composed mainly of a Delta variant, and there are two mutations present in many reads but not all (Table 4).

The first is the deletion 23,583—23,609, present in 94.73% of the reads. This mutation is interesting since, apparently, it has surpassed the wild-type population. A similar mutation is known to arise after passages in cultured cells [8], which is the case of this sample. The other mutation is G24410A, present in 70.98% of the reads.

#### 2.3.4. Sample M84

Previously, DRAGEN COVID Lineage v3.5.3 identified just the presence of Delta variant, with the mutation pattern characteristic of 21J, but this sequence did not contain three of the mutations detected by Master Mut Kit (R346K, E484K and N501Y).

Table 5 contains all AC = 3 and AC = 2 mutation groups from FreeBayes. The abundant mutations match almost all the mutations detected in the consensus sequence from DRAGEN, with one exception (G29742T mutation); This mutation was detected by DRAGEN in the consensus, but FreeBayes considered this mutation as one a part of the less abundant mutations. The number of reads for this position is 40, with a 50/50 distribution between mutant and wild-type reads. Therefore, the reason to consider this mutation in AC = 1 is because it is below the abundance threshold of FreeBayes, but it is at the abundance threshold of DRAGEN COVID Lineage v3.5.3.

In Table 6, TAAAATG28270TAAATG mutation is listed because it relates to the abundant mutation TAAAATG28270TAAATG, and Freebayes considers them as alternative alleles at the same position, and mutually exclusive. Another position also presents an alternative mutation (C23604G and C23604A), which encode the P681R and P681H mutations in the *S* gene, respectively.

Cross-contamination of the sample cannot be ruled out just by the results of screening or NGS; therefore, the sample was extracted and sequenced again, and the results were equivalent. These results can be seen in Table 7.

As we can see, there are three different assignations between both sequencing results. Mutation C4002T (First AC = 2, second AC = 3), TAAAATG28270TAAATG (First AC = 2, second AC = 3) and G29742T (First AC = 1, second AC = 2). All these changes can be originated since the percent of the mutation presence in reads is higher in the second NGS, changing from 93.96% to 98.10%, 83.01% to 84.41% and 50% to 62.92%, respectively. It is important to note that despite being at a 93.96% abundance, C4002T mutation was grouped in AC = 2; but TAAAATG28270TAAATG, at just 84.41%, is grouped in AC = 3, and the assignment of this mutation in AC = 3 groups could also be the reason of the secondary mutation at that position (TAAAATG28270TATAATG) not being listed in the vcf file of second sequencing. TGTTAA26157TA is neither listed in the vcf file since the percent of presence at the position must be higher than 15%. At last, G29868C and A29871T were not adequately covered in the first sequencing; 91.02% of the mutation detection and group assignment were fully concordant between both experiments. Five out of seven differences were due to the threshold and the assignment of groups, a larger study, with more sequencing repetitions, could help to adequately tune the threshold to an adequate value in which false negatives or positives are avoided without losing resolution.

## 3. Discussion

The Master Mut Kit showed a high concordance with NGS results and could be a valuable tool for mutation screening and variant surveillance. The mutation pattern of VOI and VOC is characteristic to them, and even if some mutations are shared, each combination represents a unique variant. Although VOC and VOI do not represent all currently circulating variants, they represent most of the cases considering the information obtained from sequencing [9]. Thus, variant identification is possible by detecting the presence or absence of specific mutations. Although this method is limited to detecting those nine mutations, the design of the test can be adapted to detect emerging variants, by introducing a new mutation pattern or by changing one or more of the currently detected mutations. Another significant drawback is the interference in the method caused by other mutations in the periphery of those detected since these changes affect the hybridization of probes and could compromise the detection [10]. Until now, these potential issues were solved by continuously updating the kit design, by including new targets and actualization of current ones.

These changes keep the kit at an update cycle, which involves the design of new assays, standardization, validation and deployment. This process is vital for developing tools to analyze highly transmissible viruses such as SARS-CoV-2.

Furthermore, the relevance of SARS-CoV-2 variants in clinical outcomes has been addressed but results aren’t homogeneous across studies [11,12]. There could be several reasons for this, from sample size, genetic background of the population, comorbidities, available medical equipment and personnel, and the method employed for variant identification. Some studies rely on sequencing to determine the variant present, others on a test, such as S-target failure. Nevertheless, the first is not available for all medical facilities, and the latter is useful just for the 69–70 deletion detection. A mutation pattern analysis can provide more information about the variant, or variants, present in a sample, than just the *S* gene dropout. Moreover, if a variant, or a specific mutation within a variant, prove to be critical in clinical outcome, symptom development, or even treatment, the identification of variant could be readily available upon SARS-CoV-2 diagnostic, even simultaneously.

Regarding the sequencing results, sample M81 is relevant because virtually all mutations are classified in group AC = 3, which means they are present in almost 100% of the reads, as the AC value indicates this, but GCT28086ACA mutation is clearly present in some reads, 150 of 376 total reads. Since this value is above the expected error rate of PCR or Illumina sequencing technology [13,14], it is possible that the analysis of mapped reads with FreeBayes reveals the rise of a new mutation from the initial population but is not visible in DRAGEN analysis since it discards them. A similar scenario is observed in sample M83, since most of the mutations are grouped within AC = 3, except for C2919T, G10097A, C26894T, C29370T and C29870A. The last two are close to the 3′, so the read number is low compared to the other sites. Setting those two sites aside, other sites have reads as high as 10,674 for the mutant base, of a total of 26,097, and they are not listed in the consensus sequence obtained from DRAGEN. Finally, for sample M86, the mutation distribution is the same between FreeBayes and DRAGEN. Within this sample, two sites have mutant and native reads, deletion at 23,583—23,609, where 94.73% of reads contain mutations, and G24410A, with 70.98% of mutant reads. The difference in percent suggests that those mutations arose at different events, and the deletion could be the first event since it has a higher relative abundance, but this should be experimentally proven. Since both percentages are higher than 50%, DRAGEN includes them at the consensus sequence; therefore, FreeBayes and this consensus sequence contain the same mutations.

As demonstrated for other viruses, could not represent a homogeneous population, but a mixture of them [15,16], and these analyses suggest that SARS-CoV-2 behaves the same way.

The result from FreeBayes analysis reflects that changes in the SARS-CoV-2 populations can be finely studied through the analysis of sequencing data as a mixture of genomes instead of a homogeneous and unique population, an application with potential for determining the genomic conservation and purity of strains. However, it is necessary to include more samples and controls to thoroughly evaluate the viability and utility of such analysis.

All three samples present a similar result between the mutations observed in DRAGEN and FreeBayes analysis, with little difference between them, but for the M84 sample the difference is higher. FreeBayes detects 78 mutations, and DRAGEN detects just 48 of them. These 30 different mutations are low abundance mutations, an abnormally high number compared with the other samples. All mutations previously detected by Master Mut Kit are listed in the FreeBayes report, with N501Y, E484K and R346K listed in the lower abundance group, which is consistent with the result of the RT-qPCR analysis of the sample.

Considering this sample as a population composed of two variants, the genomes of those hypothetical strains were determined by joining the mutation groups as follows. The first variant genome resulted from merging AC = 3 and AC = 2 groups of mutations, which is almost equal to the DRAGEN consensus genome. Furthermore, the other genome, which belongs to the less abundant, was built using AC = 3 and AC = 1 groups of mutations. The first genome was identified as a Delta variant, while the second genome was identified as a Mu variant, which is the same result obtained previously by Master Mut Kit. With this new result, Master Mut Kit analysis was fully consistent with the NGS result, but just when NGS data was analyzed with FreeBayes. The DRAGEN COVID Lineage v3.5.3 app is part of the BaseSpace platform from Illumina, an integrated online toolkit with numerous applications for a wide variety of applications. Since the diversity of analysis and the demand of computing time is that high, the deep reached at each sequencing analysis is not the best for a comprehensive analysis of fine sequencing results. The tools already available are enough for the determination of a consensus sequence, however, remain as a basic analysis resulting in the loss of essential data but analyses, such as FreeBayes, could provide more information with no experimental procedure changes. This information could be a milestone in the study of SARS-CoV-2 population dynamics or even evolution. In the future, this kind of approach can be directed to the evaluation of changes in the population originated by treatments, replacing current methods and technology and thus eliminating its limitations [17].

As stated before, data of M84 suggest the presence of both variants within the patient, but more studies must be performed to assess if both infections are active, and further, if the patient can be infected at the same time and the virus coexists, or if one of those variants dominates over the other, extinguishing it.

As shown in the tables, the analysis with DRAGEN is accurate for most of the mutations present in samples evaluated but lacks the function to detect and identify populations of genomic variants present in lower abundance. This characteristic is not part of the current epidemiological program aim, but it is important to highlight the potential data that could be obtained from this analysis. To this date, 6,160,790 submissions have been shared in the GISAID database [9]. These submissions contain not only the consensus sequence, but also their taxonomy, collection date, location and patient information, and the sequencing technology used to obtain said consensus. GISAID is designed with an epidemiological purpose, centralizing data and generating statistical analysis based on the data contributed by the whole world. Even if it contains the information of mutations present in each sample, it lacks data generated by NGS other than consensus. Sample characteristics, such as populations, mutations present in lower proportion, mutations present in the same base but in different molecules, and even simultaneous infections, are just overlooked, and the opportunity to fully characterize samples is lost. Of course, it is not an easy task to analyze and store a massive database that contains all NGS results, such as reads or mapped reads, but the storage of a record of single nucleotide mutations, insertions, and deletions in a convenient form, such as a vcf file is by far achievable in an easier way than the storage of all reads and mapping, and more convenient to analyze and compare across samples or regions.

As NGS data is composed of reads that originated from RNA amplification from the sample, it is expected that the proportion found in the sequencing data relates to the proportion present in the original sample, but this proportion can be biased in the amplification step. Nevertheless, some tools consider the percent contribution to deconvolute the reads mixture, such as MixEmt [18] which has proven the separation of haplotypes from mixtures with good accuracy [13]. Other methods such as iterative mapping against references [19] have been used to analyze closely related organisms whose genomes are mixed within a sample, or as specialized software like SNPGenie [16,19,20]. The accuracy of some tools has been analyzed, testing both human WGS and WES [21], but must be proven valid at classifying data from SARS-CoV-2 genomes. Incorporating tools like FreeBayes in NGS analysis and mutations PCR screening in common practice will increase the information available, for genomic analysis and epidemiology, respectively, and will not represent a significant difference in terms of economy, time, or specialized training.

As stated by other authors [16,17,19,20], NGS data can be exploited to obtain information further than the sequence itself, and this information can improve the understanding of the evolution of the virus, both within-host and host-to-host change, the impact of genetic drift and both natural and immunological selection, and ultimately, factors which are determinant and drive the viral genetic change over time. On the other hand, surveillance programs must be reviewed and reinforced with the deployment of new tools and algorithms in order to achieve an extensive data collection, which then could be used for evaluation of the current epidemiological situation, as well to epidemiological forecasting, and finally, enable the analysis of how these mutations arise, and disappear or become fixated, over time, not only as a biochemical and physiological event but as an epidemiological phenomenon.

## 4. Materials and Methods

### 4.1. Samples and Diagnosis

Clinical samples of nasopharyngeal and pharyngeal swabs were taken from patients with COVID-19 symptoms or people without symptoms but at risk of being infected by SARS-CoV-2. Twelve culture samples were provided by a research laboratory.

RNA extraction was performed using *Quick*-RNA™ Viral Kit (Cat. R1035, Zymo Research^®^, Irvine, CA, USA) and SARS-CoV-2 diagnosis was performed with the CoviFlu kit (Genes2Life, Irapuato, Mexico). Each RT-qPCR analysis included a positive control reaction, with a positive template included with the kit, and a negative non template reaction.

Positive samples with a threshold cycle value (Cq) of 31 or earlier were selected and analyzed with Master Mut Kit (Genes2Life, Irapuato, Mexico).

### 4.2. Sample Mutation Screening with an RT-qPCR Kit

Selected SARS-CoV-2 positive samples were analyzed with Master Mut Kit (Genes2Life, Irapuato, Mexico) to identify SARS-CoV-2 variants.

Master Mut Kit detects the following VOI and VOC key mutations within the *S* gene, in two quadruplex reactions: 69-70del, D253N, R346K, K417N, L452R/Q, T478K, E484K and N501Y. This mutation screening can also identify the Omicron variant.

This analysis was performed in either a CFX96 Touch Real-Time PCR Detection System or in CFX96 Touch Deep Well Real-Time PCR Detection System. The RT-qPCR protocol is composed of retrotranscription step (50 °C 15 m, 95 °C 2 m) and 45 cycles of amplification and fluorescence acquisition (95 °C 15 s, 58 °C 10 s, 68 °C 30 s). The fluorescence acquisition was performed at the 68 °C step through all channels. The total time of each run is around 1:40 h. Master Mut Kit result interpretation was performed with Appendix A. Each RT-qPCR analysis performed a positive control reaction, with a mutant template included in the kit, and a negative control reaction, using either NATtrol SARS-Related-Coronavirus 2 (SARS-CoV-2) Stock (ZeptoMetrix, Buffalo, NY, USA) or a sequenced native sample as template.

### 4.3. Sample Sequencing

Eighty-seven samples analyzed by Master Mut kit were further analyzed by sequencing with Illumina^®^ COVIDSeq™ Kit (Illumina, San Diego, CA, USA) in an iSeq platform, and genome sequences were obtained with the Illumina DRAGEN COVID Lineage v3.5.3 app. Samples were prepared following manufacturer instructions. PhiX Control v3 (Illumina, San Diego, CA, USA) was used in each experiment.

The resulting SARS-CoV-2 genomes were identified using the Pangolin COVID-19 Lineage Assigner web application (Available at pangolin.cog-uk.io, last accession 14 December 2021). The resulting identifications were compared to the mutations and variants previously identified by Master Mut Kit. Examples of Master Mut Kit results are presented in the Appendix A.

### 4.4. NGS Data Processing and Variant Calling

Two pathways for data analysis were followed and compared.

#### 4.4.1. Automatic Analysis: BaseSpace Sequence Hub Platform (Illumina)

The automatic data process offered by Illumina online platform was employed as the first tool. The main advantage of this online tool is the easy access and friendly user interface which have the full pipeline for SARS-CoV-2 genome sequence determination and subsequent sequence update to GISAID in one platform, thus eliminating the need to install and use each of the software programs and algorithms needed for local genome assembly; with the downside of eliminating the possibility of a deeper analysis of obtained sequencing data.

In brief, the resulting files were classified with the DRAGEN COVID Lineage v3.5.3 app. The consensus sequence obtained was compared with the reference genome of SARS-CoV-2 (NC_045512.2).

The consensus sequence was then uploaded in Nextclade (clades.nextstrain.org, last accession 14 December 2021) and the mutation list was analyzed against the results obtained from the other tools.

#### 4.4.2. Analysis with Other Bioinformatics Tools: Samtools and Freebayes

Trimmed fastq files were downloaded with BaseSpace Sequence Hub Downloader. Then, reads were mapped on the reference genome of the Wuhan SARS-CoV-2 virus (NC_045512.2) using BWA (v0.7.17-r1188). The alignments were sorted and indexed with samtools (v1.13). With this data as input, the bioinformatics tool FreeBayes (v1.3.5) was employed for variant calling. “FreeBayes can act as a frequency-based pooled caller and describe variants and haplotypes in terms of observation frequency rather than called genotypes” [22]; therefore, this tool will classify the mutations present in the fastq file depending on their relative abundance.

The resulting .bam file was analyzed with FreeBayes, with the following parameters:

freebayes -f ReferenceGenome.fna -F 0.15 -p 3 -C 10—pooled-continuous Input.bam > Output.vcf.

This command indicates that all the mutations present in at least ten reads, and representing above 15% of position depth, must be listed in the Output.vcf file. Additionally, mutations listed in the vcf file will be classified into three groups in the function of their relative abundance; those groups are low abundance (AC = 1), abundant but not dominant (AC = 2) and present in virtually all reads (AC = 3).

FreeBayes can separate the mutations in different groups because the ploidy expected from the sample can be changed. Here we used a ploidy of 3, but a different ploidy value could have a better performance depending on the sample. With this ploidy value we can separate present mutations in three clusters: The first, which is present in virtually all reads, with Spike D614G as a perfect example, and two complementary mutations sets, AC = 2 and AC = 1, each one with mutations present at lower abundance.

This means that mutation present in the higher abundance group, AC = 2, plus the mutations of AC = 3, would be from a single viral population. Therefore, mutation group AC = 1 plus mutation group AC = 3, would be the complete mutation pattern of the less abundant viral population.

The resulting vcf file is converted to a spreadsheet for data display. BAM files were visualized with Tablet [23].

## 5. Conclusions

RT-qPCR screening of mutations was fully concordant with NGS results; therefore, it can accurately measure the incidence of VOI and VOC, at a lower cost and shorter time compared to NGS. Additionally, the result obtained with this kit allowed identifying a possible co-infection case, an event hard to identify with NGS data and current bioinformatics analysis. Finally, a deeper NGS data analysis with FreeBayes vcf file, or similar software, will provide more information about the genomic characteristics of the population within a sample, and can be implemented in current databases without demanding an excessive storage capacity as it would be required for fastq o bam files.

Our results encourage the use of new validated methods which can be employed for an extensive and affordable genomic surveillance of SARS-CoV-2 variants, and recommend further development of them, especially in developing countries.

## Figures and Tables

**Table 1 ijms-23-03143-t001:** Variant identification with Master Mut Kit.

Variant	Number of Samples	Percent
Alfa	2	2.30%
Gamma	12	13.79%
Delta	34	39.08%
Epsilon/Kappa	9	10.34%
Lambda	4	4.60%
Mu	3	3.45%
P.2	11	12.64%
B.1.1.519	8	9.20%
Undetermined	4	4.60%

Undetermined samples presented no mutations or a mutation combination which did not match any of the VOI or VOC combination. Epsilon and Kappa mutants can be detected but cannot be distinguished.

**Table 2 ijms-23-03143-t002:** Mutations present in sample M81.

DRAGEN		FreeBayes						
Mutations	Insertions Deletions	Position	Reference	Mutant	Group	AO	DP	Percent
G210T		210	G	T	AC = 3	20,405	20,448	99.79%
C241T		241	C	T	AC = 3	22,996	23,134	99.40%
T1746C		1746	T	C	AC = 3	18,172	18,212	99.78%
C2061T		2061	C	T	AC = 3	78,197	78,706	99.35%
C3037T		3037	C	T	AC = 3	1871	1874	99.84%
G4181T		4181	G	T	AC = 3	41,127	41,162	99.91%
C5512T		5512	C	T	AC = 3	16,508	16,592	99.49%
C6402T		6402	C	T	AC = 3	91,214	93,221	97.85%
C7124T		7124	C	T	AC = 3	2351	2356	99.79%
C8986T		8986	C	T	AC = 3	7879	7908	99.63%
G9053T		9053	G	T	AC = 3	9440	9456	99.83%
C10029T		10,029	C	T	AC = 3	571	574	99.48%
G10642T		10,642	G	T	AC = 3	2838	2858	99.30%
A11201G		11,201	A	G	AC = 3	8884	8906	99.75%
A11332G		11,332	A	G	AC = 3	6571	6602	99.53%
C14408T		14,408	C	T	AC = 3	6106	6146	99.35%
G15451A		15,451	G	A	AC = 3	41,402	41,666	99.37%
C16466T		16,466	C	T	AC = 3	665	675	98.52%
C19220T		19,220	C	T	AC = 3	2674	2708	98.74%
G20610A		20,610	G	A	AC = 3	326	326	100.00%
C21618G		21,618	C	G	AC = 3	194	194	100.00%
C21846T		21,846	C	T	AC = 3	43	43	100.00%
A21851G		21,851	A	G	AC = 3	45	45	100.00%
G21987A		21,987	G	A	AC = 3	25	25	100.00%
	22,029–22,034	22,028	GAGTTCAG	GG	AC = 3	21	21	100.00%
T22917G		22,917	T	G	AC = 3	14,573	14,752	98.79%
C22995A		22,995	C	A	AC = 3	17,247	17,256	99.95%
A23403G		23,403	A	G	AC = 3	28,749	28,768	99.93%
C23604G		23,604	C	G	AC = 3	290	291	99.66%
G24410A		24,410	G	A	AC = 3	3414	3432	99.48%
G24872T		24,872	G	T	AC = 3	8574	8596	99.74%
G25091A		25,091	G	A	AC = 3	5441	5446	99.91%
C25469T		25,469	C	T	AC = 3	4906	4910	99.92%
T26767C		26,767	T	C	AC = 3	2788	2794	99.79%
T27638C		27,638	T	C	AC = 3	166	166	100.00%
C27752T		27,752	C	T	AC = 3	143	143	100.00%
C27874T		27,874	C	T	AC = 3	242	243	99.59%
Not detected		28,086	GCT	ACA	AC = 1	150	376	39.89%
	28,248–28,253	28,247	AGATTTCA	AA	AC = 3	28,138	28,149	99.96%
	28,271	28,270	TAAAATG	TAAATG	AC = 3	34,968	35,103	99.62%
A28461G		28,461	A	G	AC = 3	6143	6271	97.96%
G28881T		28,881	G	T	AC = 3	2135	2138	99.86%
G28916T		28,916	G	T	AC = 3	2090	2100	99.52%
G29402T		29,402	G	T	AC = 3	116	116	100.00%
G29422A		29,422	G	A	AC = 3	121	121	100.00%
C29738T		29,738	CCACG	TCACT	AC = 3	180	180	100.00%
G29742T								

AC: Group based on abundance. Three (3) is given when the mutation is present in virtually all reads, two (2) means presence in most reads, and one (1) is present in few reads. AO: Count of full observations of this alternate haplotype. DP: Total read depth at the locus. Percent: Proportion of mutant base presence concerning position depth.

**Table 3 ijms-23-03143-t003:** Mutations present in sample M83.

DRAGEN		FreeBayes						
Mutations	Insertions Deletions	Position	Reference	Mutant	Group	AO	DP	Percent
C241T		241	C	T	AC = 3	10,831	10,883	99.52%
C2919T		2919	C	T	AC = 2	9693	12,683	76.43%
C3037T		3037	C	T	AC = 3	12,503	12,518	99.88%
C4002T		4002	C	T	AC = 3	7744	7750	99.92%
C5907T		5907	C	T	AC = 3	9535	9583	99.50%
T7012G		7012	T	G	AC = 3	10,688	10,731	99.60%
C7124T		7124	C	T	AC = 3	10,827	10,838	99.90%
T7424G		7424	T	G	AC = 3	9939	9987	99.52%
C9857T		9857	C	T	AC = 3	31,512	31,684	99.46%
T9867C		9867	T	C	AC = 3	32,066	32,115	99.85%
C10029T		10,029	C	T	AC = 3	40,403	40,450	99.88%
G10097A		10,097	G	A	AC = 2	30,066	35,086	85.69%
	11,288–11,296	11,287	GTCTGGTTTTA	GA	AC = 3	33,573	33,580	99.98%
C12114T		12,114	C	T	AC = 3	17,578	18,181	96.68%
C13536T		13,536	C	T	AC = 3	24,711	24,757	99.81%
C14408T		14,408	C	T	AC = 3	16,793	16,889	99.43%
G14857T		14,857	G	T	AC = 3	8282	8299	99.80%
C19602T		19,602	C	T	AC = 3	4820	4823	99.94%
C21621G		21,621	C	G	AC = 3	8902	8913	99.88%
C21691T		21,691	C	T	AC = 3	10,200	10,212	99.88%
G21786T		21,786	GTAC	TTAT	AC = 3	8453	8492	99.54%
C21789T					AC = 3			
	22,299–22,319	22,298	AGAAGTTATTTG ACTCCTGGTGA	AA	AC = 3	482	482	100.00%
G22427C		22,427	G	C	AC = 3	2369	2375	99.75%
T22917A		22,917	T	A	AC = 3	11,399	11,438	99.66%
T23031C		23,031	T	C	AC = 3	13,148	13,162	99.89%
A23403G		23,403	A	G	AC = 3	19,068	19,081	99.93%
C23731T		23,731	C	T	AC = 3	15,612	15,643	99.80%
C24138A		24,138	C	A	AC = 3	6703	6719	99.76%
T25551C		25,551	T	C	AC = 3	12,460	12,475	99.88%
G25720T		25,720	G	T	AC = 3	21,701	21,756	99.75%
A26117T		26,117	A	T	AC = 3	15,594	15,606	99.92%
Not detected		26,894	C	T	AC = 1	10,674	26,097	40.90%
C27737T		27,737	C	T	AC = 3	7234	7235	99.99%
G27754T		27,754	G	T	AC = 3	6773	6779	99.91%
A27926G		27,926	A	G	AC = 3	8726	8735	99.90%
C28253T		28,253	C	T	AC = 3	10,600	10,622	99.79%
A28271T		28,271	A	T	AC = 3	11,873	11,914	99.66%
C28311T		28,311	C	T	AC = 3	12,321	12,409	99.29%
G28628C		28,628	G	C	AC = 3	12,507	12,521	99.89%
C28791T		28,791	C	T	AC = 3	6904	6919	99.78%
G28881A		28,881	GGG	AAC	AC = 3	6500	6541	99.37%
G28882A								
G28883C								
G28913T		28,913	G	T	AC = 3	7732	7749	99.78%
C29311T		29,311	C	T	AC = 3	4797	4812	99.69%
Not detected		29,370	C	T	AC = 1	245	1427	17.17%
	29,835	29,834	TCCCCAT	TCCCAT	AC = 3	947	951	99.58%
Not detected		29,870	C	A	AC = 1	13	54	24.07%

AC: Group based on abundance. Three (3) is given when the mutation is present in virtually all reads, two (2) means presence in most reads, and one (1) is present in few reads. AO: Count of full observations of this alternate haplotype. DP: Total read depth at the locus. Percent: Proportion of mutant base presence concerning position depth.

**Table 4 ijms-23-03143-t004:** Mutations present in sample M86.

DRAGEN		FreeBayes						
Mutations	Insertions Deletions	Position	Reference	Mutant	Group	AO	DP	Percent
G210T		210	G	T	AC = 3	1814	1820	99.67%
C241T		241	C	T	AC = 3	2097	2104	99.67%
C2061T		2061	C	T	AC = 3	7326	7368	99.43%
A2560G		2560	A	G	AC = 3	8673	8867	97.81%
C3037T		3037	C	T	AC = 3	5650	5674	99.58%
G4181T		4181	G	T	AC = 3	18,106	18,130	99.87%
C5512T		5512	C	T	AC = 3	7565	7618	99.30%
C6402T		6402	C	T	AC = 3	10,209	10,344	98.69%
C7124T		7124	C	T	AC = 3	4443	4450	99.84%
C8748T		8748	C	T	AC = 3	4748	4776	99.41%
C8986T		8986	C	T	AC = 3	3275	3286	99.67%
G9053T		9053	G	T	AC = 3	4054	4065	99.73%
C10029T		10,029	C	T	AC = 3	4583	4590	99.85%
G10642T		10,642	G	T	AC = 3	5475	5501	99.53%
A11201G		11,201	A	G	AC = 3	7780	7809	99.63%
A11332G		11,332	A	G	AC = 3	7954	7961	99.91%
C14408T		14,408	C	T	AC = 3	4662	4689	99.42%
G15451A		15,451	G	A	AC = 3	5979	6025	99.24%
C16466T		16,466	C	T	AC = 3	4340	4379	99.11%
C19220T		19,220	C	T	AC = 3	3532	3552	99.44%
C21618G		21,618	C	G	AC = 3	2031	2034	99.85%
	22,029–22,034	22,028	GAGTTCAG	GG	AC = 3	937	937	100.00%
T22917G		22,917	T	G	AC = 3	2128	2133	99.77%
C22995A		22,995	C	A	AC = 3	2745	2751	99.78%
A23403G		23,403	A	G	AC = 3	4924	4927	99.94%
	23,583–23,609	23,582	TATCAGACTCAG ACTAATTCTCCTC GGCG	TG	AC = 3	3055	3225	94.73%
G24410A		24,410	G	A	AC = 2	2307	3250	70.98%
G24872T		24,872	G	T	AC = 3	4269	4283	99.67%
G25091A		25,091	G	A	AC = 3	5328	5340	99.78%
C25469T		25,469	C	T	AC = 3	3602	3605	99.92%
T26767C		26,767	T	C	AC = 3	4811	4814	99.94%
T27638C		27,638	T	C	AC = 3	2511	2513	99.92%
C27752T		27,752	C	T	AC = 3	2183	2212	98.69%
C27874T		27,874	C	T	AC = 3	2360	2364	99.83%
G28083T		28,083	G	T	AC = 3	2157	2184	98.76%
	28,248–28,253	28,247	AGATTTCA	AA	AC = 3	3239	3241	99.94%
	28,271	28,270	TAAAATG	TAAATG	AC = 3	4552	4595	99.06%
A28461G		28,461	A	G	AC = 3	2409	2416	99.71%
G28881T		28,881	G	T	AC = 3	1130	1133	99.74%
G28916T		28,916	G	T	AC = 3	1106	1111	99.55%
G29402T		29,402	G	T	AC = 3	2800	2812	99.57%
G29422A		29,422	G	A	AC = 3	3647	3651	99.89%
G29742T		29,742	G	T	AC = 3	5034	5042	99.84%

AC: Group based on abundance. Three (3) is given when the mutation is present in virtually all reads, two (2) means presence in most reads, and one (1) is present in few reads. AO: Count of full observations of this alternate haplotype. DP: Total read depth at the locus. Percent: Proportion of mutant base presence concerning position depth.

**Table 5 ijms-23-03143-t005:** Mutations detected by Illumina DRAGEN COVID Lineage v3.5.3 and FreeBayes in sample M84.

DRAGEN		FreeBayes						
Position	Insertions Deletions	Position	Reference	Mutant	Group	AO	DP	Percent
G174A		174	G	A	AC = 2	9671	13,096	73.85%
G210T		210	G	T	AC = 2	8937	12,248	72.97%
C241T		241	C	T	AC = 3	14,162	14,229	99.53%
C2061T		2061	C	T	AC = 2	15,800	20,681	76.40%
T2974C		2974	T	C	AC = 2	6540	8770	74.57%
C3037T		3037	C	T	AC = 3	6552	6561	99.86%
G3566T		3566	G	T	AC = 2	1714	2409	71.15%
C4002T		4002	C	T	AC = 2	8666	9223	93.96%
G4181T		4181	G	T	AC = 2	23,404	29,737	78.70%
T5464G		5464	T	G	AC = 2	12,864	18,307	70.27%
C6402T		6402	C	T	AC = 2	33,362	44,293	75.32%
C6726T		6726	C	T	AC = 2	7482	10,339	72.37%
C7124T		7124	C	T	AC = 2	154	178	86.52%
C8986T		8986	C	T	AC = 2	9477	12,881	73.57%
G9053T		9053	G	T	AC = 2	12,729	17,797	71.52%
C10029T		10,029	C	T	AC = 3	6088	6092	99.93%
A11201G		11,201	A	G	AC = 2	27,328	36,591	74.69%
A11332G		11,332	A	G	AC = 2	25,626	36,350	70.50%
C14408T		14,408	C	T	AC = 3	16,780	16,865	99.50%
G15451A		15,451	G	A	AC = 2	16,857	21,728	77.58%
C16173T		16,173	C	T	AC = 2	6928	9341	74.17%
C16466T		16,466	C	T	AC = 2	2624	3663	71.64%
C16877T		16,877	C	T	AC = 2	31,851	42,991	74.09%
C19220T		19,220	C	T	AC = 2	10,815	14,387	75.17%
C21618G		21,618	C	G	AC = 2	256	340	75.29%
C21846T		21,846	C	T	AC = 3	5203	5216	99.75%
	21992:ACT	21,990	TTTATT	TTACTTCTA	AC = 2	1407	2612	53.87%
A21993C								
T21995A								
T22917G		22,917	T	G	AC = 2	3864	5170	74.74%
C22995A		22,995	C	A	AC = 2	4640	6020	77.08%
A23403G		23,403	A	G	AC = 3	19,954	19,969	99.92%
C23604G		23,604	C	G	AC = 2	7474	10,451	71.51%
C23758T		23,758	C	T	AC = 2	6368	8936	71.26%
G24410A		24,410	G	A	AC = 2	7487	9222	81.19%
G24872T		24,872	G	T	AC = 2	11,109	16,046	69.23%
C25469T		25,469	C	T	AC = 2	9879	14,243	69.36%
T26767C		26,767	T	C	AC = 2	14,953	19,483	76.75%
T27638C		27,638	T	C	AC = 2	610	789	77.31%
C27752T		27,752	C	T	AC = 2	772	1131	68.26%
C27874T		27,874	C	T	AC = 2	4605	6122	75.22%
	28,248–28,253	28,247	AGATTTCA	AA	AC = 2	11,371	13,737	82.78%
	28,271	28,270	TAAAATG	TAAATG	AC = 2	14,961	18,024	83.01%
A28461G		28,461	A	G	AC = 2	2975	4768	62.40%
G28881T		28,881	G	T	AC = 2	778	11,75	66.21%
G28916T		28,916	G	T	AC = 2	736	1099	66.97%
G29402T		29,402	G	T	AC = 2	53	86	61.63%
G29742T		29,742	G	T	AC = 1	20	40	50.00%

AC: Group based on abundance. Three (3) is given when the mutation is present in virtually all reads, two (2) means presence in most reads, and one (1) is present in few reads. AO: Count of full observations of this alternate haplotype. DP: Total read depth at the locus. Percent: Proportion of mutant base presence concerning position depth.

**Table 6 ijms-23-03143-t006:** Mutations present with fewer reads in sample M84 and detected only by FreeBayes.

Position	Reference	Mutant	Group	AO	DP	Percent
3428	A	G	AC = 1	3171	14,266	22.23%
3777	C	T	AC = 1	341	1342	25.41%
4878	C	T	AC = 1	4520	22,092	20.46%
5192	C	T	AC = 1	693	2518	27.52%
6037	C	T	AC = 1	643	2230	28.83%
6353	T	C	AC = 1	9898	46,557	21.26%
11,451	A	G	AC = 1	8578	37,592	22.82%
13,057	A	T	AC = 1	9608	42,190	22.77%
17,491	C	T	AC = 1	3394	14,388	23.59%
17,707	C	T	AC = 1	3275	13,759	23.80%
18,674	G	T	AC = 1	8815	27,825	31.68%
18,877	C	T	AC = 1	13,316	35,427	37.59%
19,035	T	C	AC = 1	5844	22,676	25.77%
20,148	C	T	AC = 1	1448	6785	21.34%
22,028	GAGTTCAG	GG	AC = 1	1193	3383	35.26%
22,599	G	A	AC = 1	1234	2852	43.27%
23,012	G	A	AC = 1	1308	5685	23.01%
23,063	A	T	AC = 1	1228	5944	20.66%
23,604	C	A	AC = 1	2972	10,451	28.44%
25,563	G	T	AC = 1	3124	12,632	24.73%
26,157	TGTTAA	TA	AC = 1	4042	21,117	19.14%
26,492	A	T	AC = 1	2116	7511	28.17%
27,616	T	C	AC = 1	169	750	22.53%
27,925	C	A	AC = 1	1885	8058	23.39%
28,005	C	T	AC = 1	2145	8278	25.91%
28,093	C	T	AC = 1	1544	8400	18.38%
28,270	TAAAATG	TATAATG	AC = 1	2993	18,024	16.61%
28,887	C	T	AC = 1	343	1174	29.22%
29,666	C	T	AC = 1	34	101	33.66%
29,779	G	T	AC = 1	13	32	40.63%

AC: Group based on abundance. Three (3) is given when the mutation is present in virtually all reads, two (2) means presence in most reads, and one (1) is present in few reads. AO: Count of full observations of this alternate haplotype. DP: Total read depth at the locus. Percent: Proportion of mutant base presence concerning position depth.

**Table 7 ijms-23-03143-t007:** Comparison between NGS results of sample M84.

Mutation Characteristics			First NGS Result		Second NGS Result		Concordance
Position	Reference	Mutant	Group	Percent	Group	Percent	
174	G	A	AC = 2	73.85%	AC = 2	69.37%	
210	G	T	AC = 2	72.97%	AC = 2	70.04%	
241	C	T	AC = 3	99.53%	AC = 3	99.54%	
2061	C	T	AC = 2	76.40%	AC = 2	76.08%	
2974	T	C	AC = 2	74.57%	AC = 2	69.02%	
3037	C	T	AC = 3	99.86%	AC = 3	99.27%	
3428	A	G	AC = 1	22.23%	AC = 1	23.10%	
3566	G	T	AC = 2	71.15%	AC = 2	67.67%	
3777	C	T	AC = 1	25.41%	AC = 1	25.60%	
4002	C	T	AC = 2	93.96%	AC = 3	98.10%	Different Group (AC) assigned
4181	G	T	AC = 2	78.70%	AC = 2	78.32%	
4878	C	T	AC = 1	20.46%	AC = 1	22.16%	
5192	C	T	AC = 1	27.52%	AC = 1	27.59%	
5464	T	G	AC = 2	70.27%	AC = 2	73.40%	
6037	C	T	AC = 1	28.83%	AC = 1	24.98%	
6353	T	C	AC = 1	21.26%	AC = 1	23.64%	
6402	C	T	AC = 2	75.32%	AC = 2	72.97%	
6726	C	T	AC = 2	72.37%	AC = 2	63.72%	
7124	C	T	AC = 2	86.52%	AC = 2	76.66%	
8986	C	T	AC = 2	73.57%	AC = 2	70.14%	
9053	G	T	AC = 2	71.52%	AC = 2	67.04%	
10,029	C	T	AC = 3	99.93%	AC = 3	99.75%	
11,201	A	G	AC = 2	74.69%	AC = 2	72.53%	
11,332	A	G	AC = 2	70.50%	AC = 2	70.63%	
11,451	A	G	AC = 1	22.82%	AC = 1	23.43%	
13,057	A	T	AC = 1	22.77%	AC = 1	23.41%	
14,408	C	T	AC = 3	99.50%	AC = 3	99.44%	
15,451	G	A	AC = 2	77.58%	AC = 2	75.14%	
16,173	C	T	AC = 2	74.17%	AC = 2	72.34%	
16,466	C	T	AC = 2	71.64%	AC = 2	72.50%	
16,877	C	T	AC = 2	74.09%	AC = 2	70.10%	
17,491	C	T	AC = 1	23.59%	AC = 1	24.36%	
17,707	C	T	AC = 1	23.80%	AC = 1	22.65%	
18,674	G	T	AC = 1	31.68%	AC = 1	31.19%	
18,877	C	T	AC = 1	37.59%	AC = 1	36.75%	
19,035	T	C	AC = 1	25.77%	AC = 1	28.78%	
19,220	C	T	AC = 2	75.17%	AC = 2	73.22%	
20,148	C	T	AC = 1	21.34%	AC = 1	23.20%	
21,618	C	G	AC = 2	75.29%	AC = 2	69.68%	
21,846	C	T	AC = 3	99.75%	AC = 3	98.98%	
21,990	TTTATT	TTACTTCTA	AC = 2	53.87%	AC = 2	61.70%	
22,028	GAGTTCAG	GG	AC = 1	35.26%	AC = 1	29.99%	
22,599	G	A	AC = 1	43.27%	AC = 1	33.21%	
22,917	T	G	AC = 2	74.74%	AC = 2	65.90%	
22,995	C	A	AC = 2	77.08%	AC = 2	66.01%	
23,012	G	A	AC = 1	23.01%	AC = 1	27.72%	
23,063	A	T	AC = 1	20.66%	AC = 1	27.76%	
23,403	A	G	AC = 3	99.92%	AC = 3	99.79%	
23,604	C	A	AC = 1	28.44%	AC = 1	28.67%	
23,604	C	G	AC = 2	71.51%	AC = 2	70.97%	
23,758	C	T	AC = 2	71.26%	AC = 2	68.60%	
24,410	G	A	AC = 2	81.19%	AC = 2	71.44%	
24,872	G	T	AC = 2	69.23%	AC = 2	66.36%	
25,469	C	T	AC = 2	69.36%	AC = 2	65.67%	
25,563	G	T	AC = 1	24.73%	AC = 1	26.06%	
26,157	TGTTAA	TA	AC = 1	19.14%	Not detected, below abundance threshold		Not detected in 2nd sequencing
26,492	A	T	AC = 1	28.17%	AC = 1	25.59%	
26,767	T	C	AC = 2	76.75%	AC = 2	73.08%	
27,616	T	C	AC = 1	22.53%	AC = 1	30.72%	
27,638	T	C	AC = 2	77.31%	AC = 2	68.52%	
27,752	C	T	AC = 2	68.26%	AC = 2	67.93%	
27,874	C	T	AC = 2	75.22%	AC = 2	53.96%	
27,925	C	A	AC = 1	23.39%	AC = 1	40.52%	
28,005	C	T	AC = 1	25.91%	AC = 1	42.64%	
28,093	C	T	AC = 1	18.38%	AC = 1	32.91%	
28,247	AGATTTCA	AA	AC = 2	82.78%	AC = 2	86.04%	
28,270	TAAAATG	TAAATG	AC = 2	83.01%	AC = 3	84.41%	Different Group (AC) assigned
28,270	TAAAATG	TATAATG	AC = 1	16.61%	Not detected, below abundance threshold		Not detected in 2nd sequencing
28,461	A	G	AC = 2	62.40%	AC = 2	57.46%	
28,881	G	T	AC = 2	66.21%	AC = 2	60.34%	
28,887	C	T	AC = 1	29.22%	AC = 1	29.52%	
28,916	G	T	AC = 2	66.97%	AC = 2	58.32%	
29,402	G	T	AC = 2	61.63%	AC = 2	79.37%	
29,666	C	T	AC = 1	33.66%	AC = 1	26.07%	
29,742	G	T	AC = 1	50.00%	AC = 2	62.92%	Different Group (AC) assigned
29,779	G	T	AC = 1	40.63%	AC = 1	28.58%	
29,868	G	C	Not Detected		AC = 2	81.82%	Not detected in first sequencing
29,871	A	T	Not Detected		AC = 1	47.02%	Not detected in first sequencing

AC: Group based on abundance. Three (3) is given when the mutation is present in virtually all reads, two (2) means presence in most reads, and one (1) is present in few reads. AO: Count of full observations of this alternate haplotype. Percent: Proportion of mutant base presence concerning position depth. In the Concordance column, a single dot (.) was used when both sequencing experiment results were the same.

## Data Availability

All data used in this article is available upon request, including fastq, bam, and vcf files.

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
