# Peer review of "An Upgrade on the Surveillance System of SARS-CoV-2: Deployment of New Methods for Genetic Inspection"

_ijms, 2022, doi:10.3390/ijms23063143_

Round 1
Reviewer 1 Report
The Authors in the Abstract state "Here, we present a method for a fast and deeper SARS-CoV-2 surveillance, from RT-qPCR screening to NGS analysis." and sentence in the Conclusions that "RT-qPCR screening of mutations was fully concordant with NGS results; therefore it 371 can be used to accurately measure the incidence of VOI and VOC. Also, the result of this 372 kit allowed the identification of a possible co-infection case, and event hard to identify 373 with NGS data and current bioinformatics analysis." The latter part of that conlusion is non fully demonstrated. Moreover, a very limited part of the paper is dedicated to the usefulness of a RT-qPCR mutation screening kit which seems, in constrast, the major aim of the study, and would require a more in-depth analysis and the lrest of the paper deals with the better performance of FreeBayes as compared with the DRAGEN software. This leads to a quite confuse presentation of the results.
The Authors state "For 86 of 87 samples (98.5%), there was full concordance between the results of the 105 Master Mut Kit analysis and the data obtained from NGS sequencing with COVIDSeq 106 Test." It should mean that the variants as indentified using NGS are the same as identified usinf Master Mut Kit analysis in 86 cases out of 87 samples. It would be more explicative insert a table listing the variants identified.
Reviewer 2 Report
he manuscript “An Upgrade on the Surveillance System of SARS-Cov-2: Deployment of New Methods for Genetic Inspection” has been reviewed.
General comments
From a public health approach, surveillance is defined as the ongoing systematic collection, analysis, and interpretation of health-related data essential to planning, implementation, and evaluation of public health practice, closely integrated with the timely dissemination of these data to those responsible for prevention and control. The goal is to provide information that can be used for health action by public health personnel, government leaders, and the public to guide public health policy and programs. Epidemiological surveillance is a multidisciplinary task that relies on several stakeholders. And because of differences in resources and system set ups, it is important to take into account the validity and precisión of the information gathered. An effective surveillance system has to be able to detect and unusual event, whether it be a new disease, a new variant or a natural disaster. This paper deals on an important tool for the effective surveillance of , in this case , SARS CoV-2 pandemic virus and on the improvement of genomic detection tools to make this information available in a more dynamic and on time manner. The work is well written yet there are some points to be improved and corrections to be made.
Specific comments
Introduction :
- Please correcte first time ACE2 and NGS, VOI and VOC ( as for all the others) are mentioned in the main text, they should be completely spelled out.
Providing a list of abreviations is recomended
- Line 78
-the amount of genomic data generated each day increase substantially
Change to : the amount of genomic data generated each day increases substantially
- Lines 86-89:
-For other viruses, the importance of this lost data has been proved: therefore a deeper analysis approaches are required to analyze the SARS-CoV-2 genome.
Change to
For other viruses, the importance of this lost data has been proven, therefore deeper analytical approaches are required to analyze the SARS-CoV-2 genome.
Provide a reference for this statement, if proven, by whom? Supportive references should be included to substantiate this claim.
- Aim of the study is lacking
Results:
Line 109
- This sentence doesn’t make sense : For determining the cause of this mismatch, we result difference????, we proceeded to deeply analyze this sample, along with 3 other samples with matching results between Master Mut Kit and COVIDSeq Test.
- In general, avoid using first person in the composition :
….we used an alternative method for this sequence assembly.
….. As we can observe…
…. As we can see,……
- Results are presented more as a discussion than plain data obtained in the analysis. Please reconsider the format of presenting results.
Discussion
Line 265 to analyzed should be changed to…. to analyze
Line 284
As stated by other authors, NGS data can be exploited to obtain information further …… Supportive references should be included to substantiate this claim.
Line 288
In the other hand change to On the oher hand
In the discussion, some concepts already reported in the introduction are repeated, so it is better to avoid unnecessary repetitions
References
Check to follow the guidelines for reference formatting
399
- Zhang, Lizhou, et al. «SARS-CoV-2 Spike-Protein D614G Mutation Increases Virion Spike Density and Infectivity». Nature 400
Communications, vol. 11, n.º 1, diciembre de 2020, p. 6013. DOI.org (Crossref), https://doi.org/10.1038/s41467-020-19808-4.
I believe this format is incorrect
Instead :
Zhang, L.; Jackson, C.B.; Mou, H.; Ojha, A.; Peng, H.; Quinlan, B.D.; Rangarajan, E.S.; Pan, A.; Vanderheiden, A.; Suthar, M.S.; Li,W.; Izard, T.; Rader, C.; Farzan, M.; Choe, H. SARS-CoV-2 spike-protein D614G mutation increases virion spike density and infectivity. Nat. Commun. 2020,11, 6013. doi: 10.1038/s41467-020-19808-4.
Spacing, punctuation marks, grammar, and spelling errors should be reviewed thoroughly.
Round 2
Reviewer 1 Report
Suitable for publication